# S100A4-lineage cells contribute modestly to angiotensin II-mediated thoracic aortic aneurysms through angiotensin II type 1a receptor in mice

Sohei Ito[1,2], Jeff Z. Chen[1,3], Deborah A. Howatt[1], Jessica J. Moorleghen[1], Hong S. Lu[1,2,3], Alan Daugherty[1,2,3]*, Hisashi Sawada[1,2,3,4,5]*

1 Saha Cardiovascular Research Center, University of Kentucky, Kentucky, United States of America,
2 Saha Aortic Center, University of Kentucky, Kentucky, United States of America, 3 Department of Physiology, University of Kentucky, Kentucky, United States of America, 4 Tsukuba Institute for Advanced Research, University of Tsukuba, Ibaraki, Japan, 5 Life Science Center for Survival Dynamics, Tsukuba Advanced Research Alliance, University of Tsukuba, Ibaraki, Japan

* hisashi.sawada@uky.edu (HS); Alan.Daugherty@uky.edu (AD)

## Abstract

Angiotensin II (AngII) exerts a critical role in thoracic aortic aneurysm (TAA) formation via AngII type 1a receptor (AT1aR). However, the principal cell type mediating this process remains unclear. Our previous study demonstrated that S100A4-lineage cells are present in the aortic wall and involved in AngII-induced vascular remodeling. In the present study, we investigated whether S100A4-lineage cells contribute to AngII-mediated TAA formation through AT1aR. Proteomic, bulk RNA sequencing, and single-cell RNA sequencing data were analyzed to assess changes in S100A4 abundance in response to AngII infusion. Lineage tracing was performed to track S100A4-lineage cells during AngII-mediated TAA formation. Either saline or AngII was infused in mice with genetic deletion of AT1aR in S100A4-lineage cells and their wild-type littermates. AngII infusion increased S100A4 protein and mRNA abundance significantly in the ascending aorta, particularly within smooth muscle cells and fibroblasts. Lineage tracing revealed that S100A4-positive cells were localized to the media and adventitia under basal conditions. Following AngII infusion, S100A4-lineage cells expanded markedly throughout the entire aortic wall and comprised a heterogeneous population including smooth muscle cells and fibroblasts. Deletion of AT1aR in S100A4-lineage cells partially reduced AngII-induced TAA formation. In conclusion, S100A4-lineage cells modestly contribute to AngII-mediated TAA development through AT1aR in mice.

**Data availability statement:** All numerical data are available in Supplemental Excel File. Raw bulk RNA sequencing and scRNAseq data are available at the Gene Expression Omnibus (GSE309042, GSE275132).

**Funding:** The studies reported in this manuscript were supported by the National Heart, Lung, and Blood Institute of the National Institutes of Health (R35HL155649, R01HL133723 [AD]), the American Heart Association (23MERIT1036341 [AD], 24CDA1268148 [HS]), the Leducq Foundation for the Networks of Excellence Program (Cellular and Molecular Drivers of Acute Aortic Dissections; 22CVD03 [AD]), and EXcellence Program for Engaging Research Talent - Japan (EXPERT-J, JPMJEX2506 [HS]). There was no additional external funding received for this study. The funders had no role in study design, data collection and analysis, decision to publish, or preparation of the manuscript.

**Competing interests:** The authors have declared that no competing interests exist.

## Introduction

Thoracic aortic aneurysm (TAA) is a serious vascular disease associated with a high risk of life-threatening events, including aortic dissection and rupture [1,2]. Angiotensin II (AngII), a central effector peptide of the renin-angiotensin system, exerts its biological effects primarily through stimulation of AngII type 1 receptors (AT1R in humans and AT1aR and AT1bR in mice) [3]. Subcutaneous infusion of AngII induces TAA in mice [4,5]. Pharmacological blockade and genetic deletion of AT1aR significantly suppress TAA development [6,7]. These findings provide compelling evidence suggesting the critical role of AT1aR in AngII-mediated TAA formation. However, the principal cell type mediating AT1aR-dependent effects in AngII-mediated TAA remains unknown.

The aortic wall is composed of three distinct layers: intima, media, and adventitia [8]. These layers are primarily composed of endothelial cells, smooth muscle cells (SMCs), and fibroblasts, respectively. Resident macrophages are also present within the aortic wall. A number of studies have demonstrated that all of these cell types may contribute to the pathogenesis of TAA through mechanisms such as endothelial dysfunction, SMC phenotypic modulation, adventitial remodeling, and inflammation [9]. Despite these contributions, our previous studies have shown that cell type-specific deletion of AT1aR in endothelial cells, SMCs, or myeloid cells exerted no or modest effects on AngII-induced TAA [6]. These findings suggest that deletion in each cell type alone is insufficient to limit disease development. Given that fibroblasts are one of the major cell types in the aortic wall, AT1aR in fibroblasts may contribute to AngII-induced TAA formation. However, preclinical studies to investigate the role of AT1aR in fibroblasts have been restricted by the lack of specific Cre drivers to selectively target this cell type in vivo.

S100A4 is a calcium-binding protein, known for its significant role in fibrosis across multiple organs [10–13]. S100A4 is also known as fibroblast-specific protein 1. In addition to its role in organ fibrosis, S100A4 is involved in atherosclerosis in mice and humans through promoting SMC phenotypic switching and amplifying inflammatory responses [14,15]. Our previous study demonstrated that S100A4-lineage cells are observed in the aortic adventitia and contribute to AngII-induced vascular remodeling [16]. However, our lineage tracing studies revealed marked proliferation and expansion of S100A4-lineage cells in both the media and adventitia of the ascending aorta following AngII infusion, coinciding with medial and adventitial thickening, that are hallmarks of TAA formation. Immunostaining showed that S100A4-lineage cells were not restricted to fibroblasts but were also present in proximity to vascular smooth muscle cells and endothelial cells. These findings indicate that S100A4 identifies a heterogeneous population of cells rather than just fibroblasts, as originally defined for this promoter [17]. Accordingly, in this study, we focused on the functional role of this heterogeneous population of S100A4-lineage cells in AngII-induced TAA formation. Therefore, we hypothesized that S100A4-lineage cells contribute to AngII-induced TAA formation through AT1aR signaling. In this study, we investigated the role of S100A4-lineage cells in AngII-induced TAA by multi-omics profiling, lineage tracing, and cell type-specific deletion of AT1aR.

## Materials and methods

### Mice

AT1a receptor floxed mice were developed by InGenious Targeting, Inc. (Holbrook, NY) and we donated this mouse strain to The Jackson Laboratory (#016211, Bar Harbor, ME) [6]. C57BL/6J (#000664), S100A4-Cre (#012641) [16,18,19], ROSA26R$^{LacZ}$ (#003474), and low-density lipoprotein receptor-deficient (LDLR-/-, #002077) mice were purchased from The Jackson Laboratory. Mice were bred to generate S100A4-Cre 0/0 or +/0 ROSA26R$^{LacZ}$+/0 mice and S100A4-Cre 0/0 or +/0 AT1aR floxed LDLR-/- mice. Due to the lack of antibodies that authentically detect AT1aR, direct validation of protein deletion was not feasible [6,20,21]. Recombination efficiency was assessed by lineage tracing. Male mice were used due to their greater susceptibility to AngII-induced aneurysm formation [22]. Mice were housed in ventilated cages with negative air pressure (Allentown Inc., Allentown, NJ). Drinking water filtered by reverse osmosis and a normal diet (#2918, Inotive, West Lafayette, IN) were provided ad libitum. At 8–10 weeks of age, LDLR-/- mice were fed a Western-type diet for 12 weeks (#TD.88137, Harlan Teklad, Indianapolis, IN). Rooms were set with light/dark cycles (14 and 10 hours), temperature (20−23°C), and humidity (50−60%). All surgical procedures were performed under anesthesia induced with isoflurane. Mice were euthanized by overdose of ketamine/xylazine cocktail. All efforts were made to minimize animal suffering, including careful monitoring of animal health, adherence to humane endpoints, and use of appropriate anesthesia during all procedures. The studies followed the recommendations of The Guide for the Care and Use of Laboratory Animals (National Institutes of Health). All procedures were approved by the University of Kentucky Institutional Animal Care and Use Committee (Protocol # 2006−0009).

### Osmotic pump implantation

Either saline or AngII (1,000 ng/kg/min, #H-1705, Bachem Americas, Inc., Torrance, CA) was infused for 3 or 28 days using mini-osmotic pumps (Model 2001 or 2004, Alzet LLC, Campbell, CA). The pumps were implanted subcutaneously on the right back of mice, as described previously [23]. Mice were randomly assigned to study groups.

### Tissue processing for β-gal staining

Adrenal gland, brain, kidney, liver, lung, skin, spleen, thymus, and thoracic aorta were harvested from S100A4-Cre 0/0 and +/0 ROSA26R$^{LacZ}$+/0 mice. The thoracic aorta was also harvested from S100A4-Cre+/0 ROSA26R$^{LacZ}$+/0 mice after 28 days of either saline or AngII infusion. Mice were terminated by an overdose of ketamine/xylazine mixture (90 mg/kg and 10 mg/kg, respectively). The right atrium was cut for exsanguination and perfusate drainage and saline (8–10 mL) was perfused through the left ventricle. Tissues were dissected free and immersed in paraformaldehyde (PFA, 4% wt/vol) for 1 hour at 4°C or neutral buffered formalin (10% wt/vol) overnight at room temperature.

### β-gal staining

β-gal staining was performed using the protocol described previously [24]. Tissues were incubated in buffer containing sodium phosphate (100 mM, pH 7.3), MgCl$_2$ (2 mM), sodium deoxycholate (0.01% wt/vol), and NP40 (0.02% wt/vol) for 90 minutes. Subsequently, X-gal (1 mg/mL, V394A, Promega), potassium ferricyanide (5 mM), and potassium ferrocyanide (5 mM) were added to the buffer, and samples were incubated overnight at room temperature. Whole tissues were then post-fixed with buffered formalin (10% wt/vol). Tissue images were captured using a dissection microscope with a high-resolution camera (DS-Ri1, Nikon). Thoracic aortas were then cut into 10 µm sections, stained with eosin (1% wt/vol) for 2 minutes, and coverslipped using glycerol gelatin (GG1, MilliporeSigma). Cross-sectional aortic images were captured using a microscope (E600, Nikon) with a high-resolution camera.

### Immunostaining with β-gal staining

Immunostaining for α-smooth muscle actin (αSMA) and ER-TR7 was performed after β-gal staining. Briefly, β-gal-stained slides were incubated with H$_2$O$_2$ and avidin/biotin blocker (SP-2001, Vector Laboratories, Burlingame, CA). The slides

were then incubated with either αSMA (20 μg/mL, ab5694, Abcam, Cambridge, MA) or ER-TR7 (1 μg/mL, ab51824, Abcam) for 15 minutes at 40°C followed by biotinylated anti-rabbit (1:500, BA-1000, Vector Laboratories) and anti-rat IgG antibody (BA-4001, Vector Laboratories), respectively. An Elite horseradish peroxidase ABC kit (PK-6100; Vector Laboratories) and AEC chromogen (ImmPACT AEC, SK-4205; Vector Laboratories) were used to visualize reactivity.

## Plasma renin concentrations

Blood (500 μL) was collected by cardiac puncture (right ventricle) into a syringe containing 10 μL of 0.5 M EDTA. Plasma renin concentrations were measured using mouse renin ELISA kits (DY4277, R&D Systems, Minneapolis, MN) in plasma samples supplemented with exogenous mouse recombinant AGT, as described previously [25].

## Systolic blood pressure measurements

Systolic blood pressure was measured by a non-invasive tail cuff system (Coda 8, Kent Scientific), as described previously [26]. Mice were restrained in a holder and placed on a heated platform. Blood pressure was measured 20 times at the same time each day for three consecutive days. Data showing <60 or >250 mmHg, standard deviation >30 mmHg, or collected cycles <5 of 20 were excluded.

## Quantification of TAA

Fixed thoracic aortas were transferred to saline. After removing the adventitia using forceps, the aortas were opened en face, pinned, and photographed. Ascending aortic area was measured using ImagePro Plus software (Media Cybernetics, Inc., Bethesda, MD), as described previously [5,27].

## Bulk RNA sequencing

Ascending aortas were harvested from male C57BL/6J mice infused with either saline or AngII (1,000 ng/kg/min) for 3 days (n = 12 per group). Periaortic tissues and endothelial cells were removed. Ascending aortas displaying intramural hemorrhage were excluded. Two aortic samples of each group were pooled as one sample for RNA sequencing. The pooled samples were then incubated with RNAlater solution (#AM7020, Invitrogen, Carlsbad, CA) overnight. Subsequently, mRNA was extracted using RNeasy Fibrous Tissue Mini kits (#74704, Qiagen, Hilden, Germany) and shipped to Novogene (Sacramento, CA) for mRNA sequencing (n = 6 biological replicates per group).

A sequencing library was generated from total mRNA (1 μg) using NEBNext UltraTM RNA Library Prep Kits for Illumina (San Diego, CA). cDNA libraries were sequenced by a NovaSeq 6000 (Illumina) in a paired-end fashion to reach more than 1,500,000 reads. FASTQ sequence data were mapped to the mouse genome mm10 using STAR (v2.5, mismatch = 2) and quantified using HTSeq (v0.6.1, -m union).

## Statistical analysis

Data, except for single-cell RNA sequencing (scRNAseq) results, are represented as individual data points with the median and 25th/75th percentiles. Normality and homogeneity of variance were assessed by the Shapiro-Wilk and Brown-Forsythe tests, respectively. TMM-normalized read count, protein intensity data, and number of β-gal positive cells were analyzed by Student's t-test. For datasets involving four groups, two-way ANOVA followed by Holm–Sidak multiple comparison correction was applied. P < 0.05 was considered statistically significant. Statistical analyses were performed using SigmaPlot version 15.0 (SYSTAT Software Inc., San Jose, CA).

scRNAseq data were downloaded from the Gene Expression Omnibus (GSE275132). Seurat package (v4.3.0 or 4.3.3) and R (v4.1.0 or 4.3.2) were used to analyze scRNAseq data in R Studio (2022.12.0) [28,29]. Seurat objects for each of the UMI count datasets were built using the "CreateSeuratObject" function by the following criteria: ≥3 cells and ≥200

detected genes. Cells expressing fewer than 200 or more than 5,000 genes were filtered out for exclusion of non-cell or cell aggregates, respectively. Cells with more than 10% mitochondrial genes were also excluded. UMI counts were then normalized as follows: counts for each cell were divided by total counts, multiplied by 10,000, and transformed to a natural log. "FindIntegrationAnchors" and "IntegrateData" functions were used to remove batch effects and integrate the four normalized datasets. Uniform manifold approximation and projection (UMAP) dimensionality reduction using the first 20 principal components (PCs) was applied, and cell clustering was performed using a shared nearest neighbor approach with a resolution parameter of 0.5. "FindAllMarkers" and "FindConcervedMarkers" functions were used to identify conserved marker genes to determine cell types of each of the clusters. Cell clusters were annotated based on canonical lineage markers. Smooth muscle cells were identified by expression of *Myh11*, *Acta2*, and *Tagln*, endothelial cells by *Pecam1* and *Cdh5*, fibroblasts by *Dcn*, *Lum*, and *Col1a1*, and macrophages by *Lyz2* and *Cd68*. These marker genes were used to define the major cell populations in UMAP projections and downstream analyses. The Wilcoxon rank-sum test implemented in the FindMarkers function of Seurat (v4.3.3) was used to identify differentially expressed genes in S100a4-positive cells from the ascending aortas of control and AngII-infused mice. False discovery rate (FDR) was calculated using the Benjamini–Hochberg method. Genes with $FDR < 0.05$, $|log_2FC| > 0.25$, and detected in ≥10% of cells in either group were considered significantly differentially expressed.

## Results

### Aortic S100A4 abundance was increased prior to AngII-induced TAA formation

First, we analyzed multi-omics datasets to investigate how S100A4 abundance changes in response to AngII infusion but prior to overt TAA formation (Fig 1A). Proteomics data were obtained from our previous study in which mice were infused with either saline or AngII for 3 days. Mice with gross ascending aortic pathology were excluded. scRNAseq data were also obtained from our previous study in which ascending aortas were harvested at either baseline or day 3 of AngII infusion. In addition, we performed bulk RNA sequencing using ascending aortas from mice also subjected to 3 days of either saline or AngII infusion. The proteomics and bulk RNA sequencing demonstrated that S100A4 protein and mRNA abundance were increased by 3 days AngII infusion before the onset of TAA (Fig 1B, 1C). In scRNAseq, UMAP visualization showed prominent *S100a4* mRNA abundance not only in fibroblasts (FBs) but also smooth muscle cells (SMCs), with violin plots confirming increased *S100a4* transcripts in these populations upon AngII stimulation (Fig 1D, 1E). The scRNAseq data were further analyzed to investigate the involvement of S100A4 in AngII-induced TAA formation. Aortic cells were stratified based on *S100a4* mRNA abundance, and those expressing *S100a4* were extracted as *S100a4*-positive cells (S2 Fig A). Differentially expressed genes between control and AngII-infused samples within *S100a4*-positive cells were then identified (S2 Fig B). Gene ontology enrichment analyses revealed that *S100a4*-positive cells exhibited upregulation of genes associated with mitochondrial transcription and translation, cellular respiration, and metabolic processes following AngII infusion (S2 Fig C, D).

### S100A4-lineage cells were present in multiple fibrous organs and increased in the aorta following AngII infusion

To determine the tissue distribution of S100A4-lineage cells, lineage tracing experiments were performed using ROSA26R[LacZ] reporter mice. Homozygous ROSA26R[LacZ] +/+ mice were bred with S100A4-Cre +/0 mice to generate S100A4-Cre 0/0 or +/0 ROSA26R[LacZ] +/0 mice (Fig 2A). Major organs, the aorta, brain, kidney, liver, lung, skin, and thymus, were harvested at 10 weeks of age. No β-gal–positive signal was detected in any organs of S100A4-Cre 0/0 mice (Fig 2B). While only minimal labeling was observed in non-fibrous organs, such as the brain, kidney, and liver, β-gal staining revealed widespread distribution of S100A4-lineage cells in fibrous organs, lung, skin, thymus, and aorta of S100A4-Cre +/0 mice (Fig 2B). Histological analyses further validated that β-gal staining was detected only in S100A4-Cre +/0 mice, but not in S100A4-Cre 0/0 controls, supporting the specificity of the staining (S1 Fig).

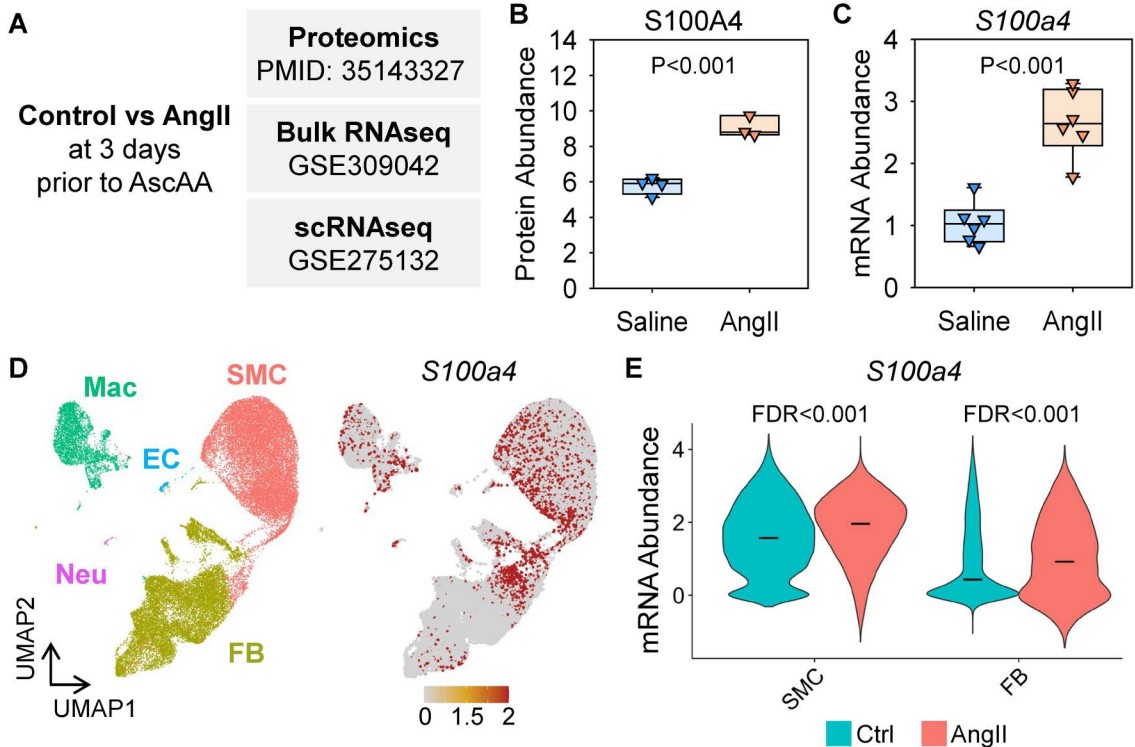

**Fig 1. Multi-omics analyses identified increased aortic S100A4 abundance after 3 days of AngII infusion in mice. (A)** S100A4 abundance was determined in 3 types of omics analyses: mass-spectrometry-assisted proteomics, bulk RNAseq, and scRNAseq. **(B)** Proteomics and **(C)** bulk RNAseq for S100A4 in the ascending aorta of mice infused with either saline or AngII for 3 days. **(D)** Uniform manifold approximation and projection (UMAP) plots of all aortic cells at baseline and after 3 days of AngII infusion. SMC indicates smooth muscle cell; FB, fibroblast; EC, endothelial cell; Mac, macrophage; Neu, neural cell. **(E)** Violin plots for *S100a4* mRNA in SMCs and FBs. Black bars indicate the median.

## AngII infusion increased cells in the media and adventitia with high beta-galactosidase activity in S100A4-Cre x ROSA26R^LacZ mice

Subsequently, we investigated the impact of AngII infusion on the distribution of S100A4-lineage cells during TAA formation. β-gal staining was performed on aortic tissue from S100A4-Cre+/0 ROSA26R^LacZ+/0 mice infused with either saline or AngII for 28 days. The intensity of β-gal-positive areas in the thoracic aorta of AngII-infused mice increased compared to saline-infused mice (Fig 3A). Cross-sectional images revealed that S100A4-lineage cells were distributed sparsely in the ascending aorta of saline-infused mice, whereas they were distributed more widely in AngII-infused mice (Fig 3B). S100A4-lineage cells were observed in the intima, media, and adventitia of AngII-infused mice (Fig 3B), and the number of β-gal-positive cells was increased significantly in AngII-infused mice (Fig 3C). Immunostaining demonstrated co-localization of β-gal with α-smooth muscle actin (α-SMA) and ER-TR7, suggesting that the cell types of S100A4-lineage cells are SMCs and fibroblasts in the aortic wall (Fig 3D).

## Deletion of AT1aR in S100A4-lineage cells modestly attenuated AngII-induced TAA formation

To determine the role of AT1aR in S100A4-lineage cells in AngII-mediated TAA formation, S100A4-lineage cell-specific AT1aR-deficient (S100A4-AT1aR-/-) mice on a LDLR-/- background were generated.

Either saline or AngII was infused subcutaneously into S100A4-AT1aR-/- mice and their wild-type littermates (S100A4-AT1aR+/+). AngII infusion increased systolic blood pressure significantly and decreased plasma renin concentrations

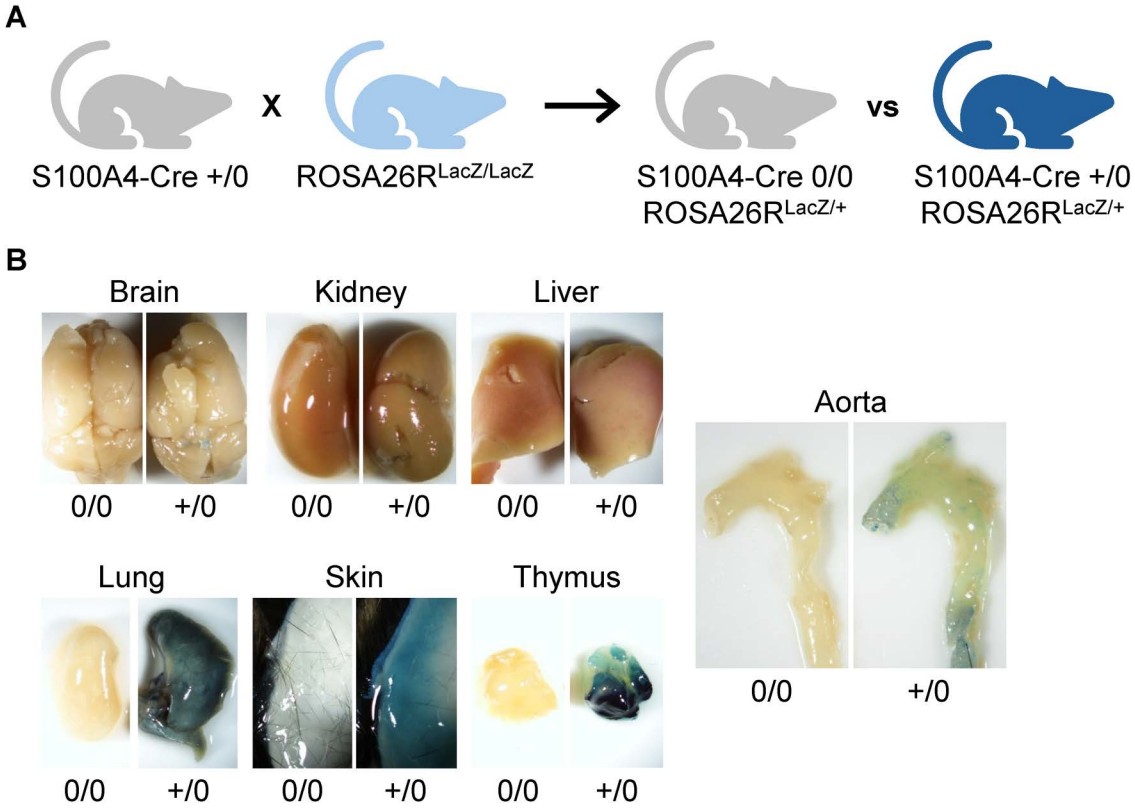

**Fig 2. Distribution of S100A4-lineage cells in major organs. (A)** Breeding strategy for S100A4-Cre+/0 ROSA26R$^{LacZ}$+/0 mice. **(B)** Representative images of β-gal staining in major organs of male S100A4-Cre 0/0 (left) and +/0 (right) ROSA26R$^{LacZ}$+/0 mice.

regardless of genotype (Fig 4A, 4B), validating effective delivery of AngII. No deaths occurred in saline-infused mice of either genotype. Thoracic aortic rupture was observed in 16–18% of AngII-infused mice in both groups (wild-type: 5/28; deficient: 5/32), with no significant difference between genotypes (P = 0.99, Fisher's exact test). As expected, AngII infusion increased thoracic aortic areas in S100A4-AT1aR+/+ mice, indicating TAA development (Fig 5A, 5B). Of note, S100A4-AT1aR-/- mice exhibited a modest reduction in aortic dilatation compared to their wild-type littermates (Fig 5A, 5B). These results indicate that AT1aR signaling in S100A4-lineage cells contributed partially to AngII-induced aortic aneurysm formation and does not affect atherosclerosis development.

## Validation of AT1aR deletion in S100A4-lineage cells

Since no reliable AT1aR antibodies that authentically detect the protein are available, it was not feasible to directly validate the efficiency of AT1aR deletion in these cells by immunostaining or Western blotting. However, the presence of Cre and floxed alleles was confirmed by DNA PCR using tail tissues (data not shown), and S100A4-Cre-mediated recombination activity was validated using the ROSA26R$^{LacZ}$ reporter system (Fig 2), supporting the effectiveness of AT1aR deletion.

## Discussion

In this study, we found that S100A4 abundance increased in the thoracic aorta prior to overt AngII-induced aortopathy, as demonstrated by multi-omics analyses. Lineage tracing revealed that S100A4-lineage cells were distributed broadly in fibrous tissues and expanded in the aortic wall upon AngII infusion. These cells were localized to the media and adventitia and

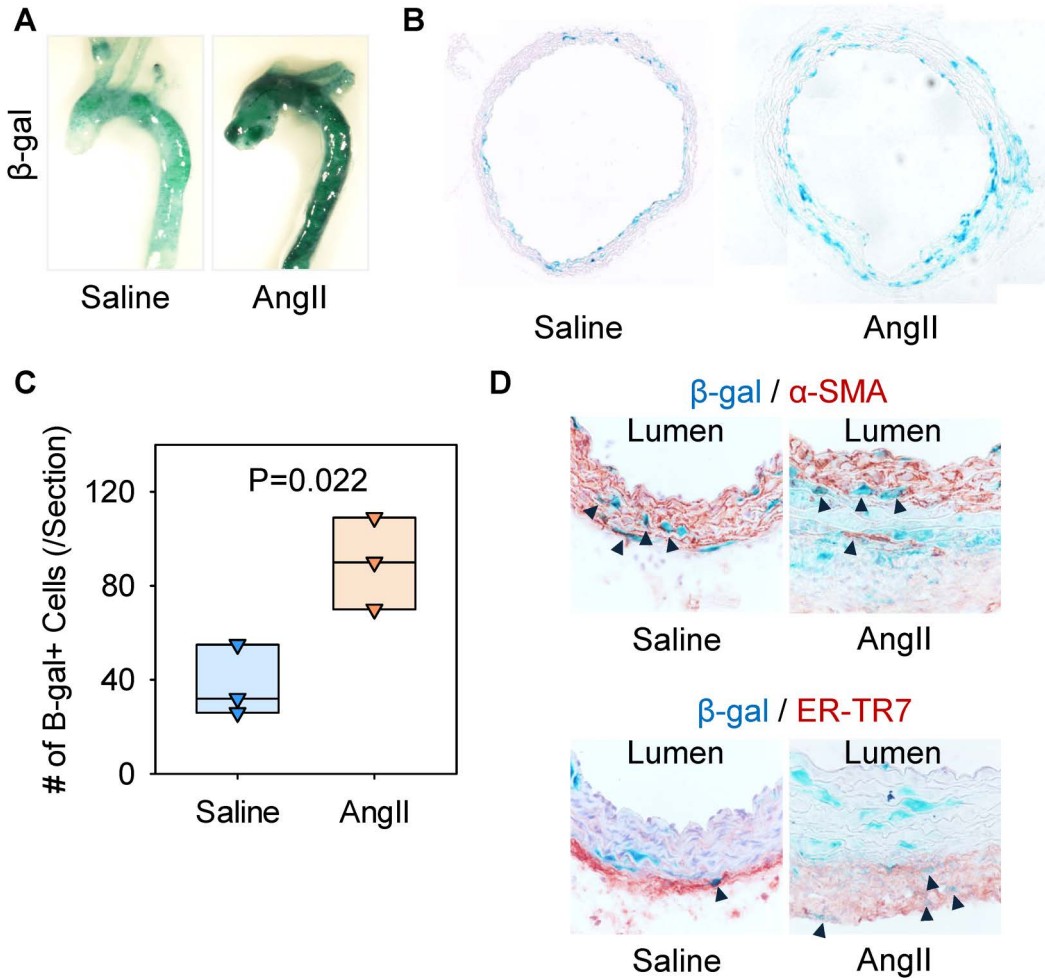

**Fig 3. AngII infusion increased S100A4-lineage cells in the ascending aorta of mice.** Since no β-gal signal was detected in S100A4-Cre-negative control mice (S1 Fig), only S100A4-Cre-positive mice are shown and compared following either saline or AngII infusion. Representative images of β-gal staining in **(A)** ex vivo of the thoracic aorta and **(B)** cross-sections of the ascending aorta in S100A4-Cre+/0 ROSA26R^LacZ+/0 male mice with either saline or AngII infusion for 28 days. N = 3/group. **(C)** Number of β-gal positive cells. P-value was determined by Student's t-test. **(D)** Representative images of β-gal staining followed by immunostaining for α-SMA and ER-TR7. Black arrows indicate representative β-gal positive cells co-localizing with α-SMA or ER-TR7 in the aortic wall.

included both fibroblasts and SMCs. These observations indicate that S100A4 identifies a heterogeneous population of cells rather than a single defined lineage. Importantly, deletion of AT1aR in S100A4-lineage cells attenuated AngII-induced TAA formation modestly. These findings suggest a partial contribution of S100A4-lineage cells to AngII-mediated TAA development.

We demonstrated that AngII infusion increased S100A4 in the ascending aorta. The transcriptional regulation of S100A4 is linked to multiple signaling pathways and transcription factors, including Smad, Wnt, and hypoxia-inducible factors [30–32]. AngII-stimulated AT1aR signaling may influence S100A4 expression through these regulatory mechanisms. However, the precise transcriptional and epigenetic control of S100A4 in aortic cells remains to be determined, and further studies are required to clarify these mechanisms.

In the present study, deletion of AT1aR in S100A4-lineage cells led to a partial reduction in AngII-induced thoracic aortic dilatation and had no effect on the incidence of aortic rupture. Since the efficiency of AT1aR deletion in S100A4-lineage

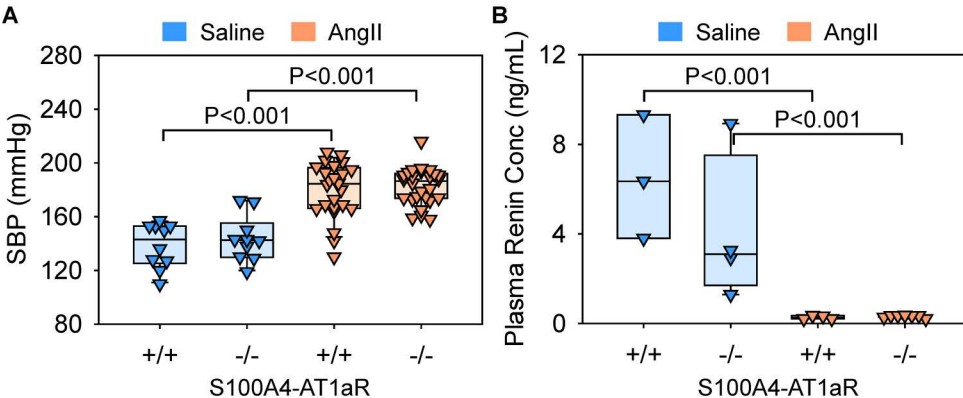

**Fig 4. Determination of AngII infusion in S100A4-specific AT1aR-deficient mice and their littermates. (A)** Systolic blood pressure and **(B)** plasma renin concentrations in saline- or AngII-infused male LDLR-/- mice. N = 10-27/group. P-values were determined by two-way ANOVA followed by Holm-Sidak test.

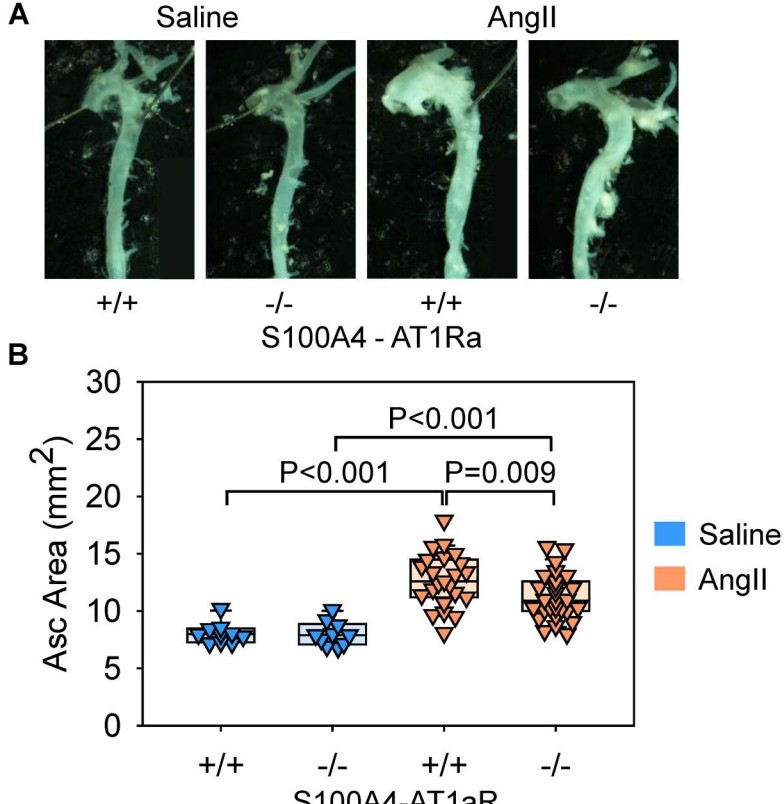

**Fig 5. AT1aR deletion in S100A4-lineage cells attenuated AngII-induced thoracic aortic aneurysm. (A)** Representative ex vivo images of thoracic aorta and **(B)** ascending aortic areas measured by the en face method in saline- or AngII-infused male LDLR-/- mice. N = 10-27/group. P-values were determined by two-way ANOVA followed by Holm-Sidak test.

cells could not be directly validated, the modest phenotype could potentially reflect incomplete protein deletion. None-theless, these findings suggest that S100A4-lineage cells are not the sole mediators of AT1aR-dependent effects in TAA. S100A4-lineage cells reside in the adventitia, but the S100A4-Cre system does not target all fibroblasts in the aortic wall [16]. Thus, fibroblasts that are not derived from the S100A4 lineage (non-S100A4-lineage fibroblasts) may play a signifi-cant role in disease progression. The adventitia contains diverse fibroblast subpopulations with distinct developmental ori-gins [33,34]. In the ascending aorta, which is the prone region of AngII-mediated TAA, our previous studies demonstrated that the cardiac neural crest and second heart field are the embryonic origin of adventitial cells [24,35,36]. In addition, there are fibroblasts expressing stem cell antigen 1, Sca1, a marker of progenitor cells, in the adventitia [37–40]. It is plau-sible that certain fibroblast subsets outside the S100A4 lineage contribute to aortic pathology through AT1aR-dependent mechanisms. Further studies employing more specific genetic tools and single-cell-based approaches will be essential to delineate the role of non-S100A4-lineage fibroblasts in AngII-induced TAA.

There is compelling evidence that SMC phenotypic transition, by which contractile SMCs change to a synthetic and inflammatory phenotype, has been recognized as a key mechanism of medial disruption in TAA [41]. It has also been reported that macrophage accumulation in the adventitia is a prominent feature of TAA [9,42]. In the present study, lineage-tracing experiments revealed that S100A4-lineage cells were expanded within both the media and adventitia by AngII infusion. Therefore, although their contribution appears to be limited, S100A4-lineage cells may promote AngII-mediated TAA through medial disruption and adventitial inflammation. Our scRNAseq analysis revealed that AngII infusion significantly altered mitochondrial-related mRNA abundance in S100a4-positive cells. These findings suggest that AngII modulates mitochondrial function and metabolic regulation in S100a4-positive cells. This transcriptional signature is consistent with a previous study implicating S100A4 in the regulation of mitochondrial function and cellular metabolism [43]. The previous study has shown that S100A4 upregulates the mitochondrial complex I subunit NDUFS2, thereby enhancing oxidative phosphorylation. In addition, S100A4 has been implicated in multiple mitochondrial-dependent processes, including the regulation of mitophagy, the response to oxidative stress, and metabolic reprogramming [44,45]. Increasing evidence indicates that mitochondrial dysfunction contributes to aortic aneurysm development [46–48]. Thus, it is plausible that mitochondria-associated pathways in S100A4-lineage cells may contribute to the modest effects observed in the present study. Further investigation is needed to elucidate the molecular mechanisms and cellular interactions through which S100A4-lineage cells mediate these pathologies.

S100A4 has also been named as the fibroblast-specific protein 1 [49]. Therefore, in our previous study, S100A4-Cre was used initially as a fibroblast-specific driver [16]. However, we found that S100A4-lineage cells were present not only in the adventitia but also in the media, which is primarily composed of SMCs. Therefore, the use of S100A4-Cre mice has limitations for specifically investigating the role of fibroblasts. Although multiple Cre driver lines are available to track fibroblasts in mice [50], relatively few studies have applied them to vascular research, including investigations of the aorta. Among these, TCF21-CreERT2 is considered relatively specific for adventitial fibroblasts and has been used to trace fibroblast-to-SMC transitions [51,52]. However, its expression can be dynamically regulated during vascular remodeling, complicating the determination of the role of fibroblasts. Col1a2-CreERT2 targets collagen-producing fibroblasts but also labels other cells, including SMCs [53,54]. PDGFRα-CreERT2 offers relatively high specificity for adventitial fibroblasts and has minimal overlap with medial SMCs [55]. While PDGFRα is also expressed in other mesenchymal populations during development [56], it largely localizes to adventitial fibroblasts in the adult aorta. The previous study demonstrated that PDGFRα-lineage cells contribute to vascular remodeling [55]. Thus, future studies should further dissect the roles of distinct fibroblast subsets within S100A4-lineage cells using more cell-type-specific genetic tools, such as PDGFRα-CreERT2 in combination with AT1aR deletion.

In conclusion, S100A4-lineage cells expand in the thoracic aortic wall in response to AngII infusion and contribute par-tially to AngII-mediated TAA formation in mice.

## Supporting information

**S1 Fig. Distribution of S100A4-lineage cells in fibrous organs.** Representative histological images of β-gal staining in the lung, skin, thymus, and aorta of male S100A4-Cre 0/0 (left) and +/0 (right) ROSA26R$^{LacZ}$+/0 mice.
(TIF)

**S2 Fig. AngII-induced transcriptomic alterations in *S100a4*-positive cells. (A)** Identification of *S100a4*-positive cells based on *S100a4* mRNA read counts. **(B)** Volcano plot, **(C)** gene ontology (GO) enrichment circle plot, and **(D)** GO enrichment chord plot generated from differentially expressed genes in *S100a4*-positive cells of the ascending aorta between control and AngII-infused mice.
(TIF)

**S1 File. Data Summary.**
(XLSX)

## Author contributions

**Conceptualization:** Alan Daugherty, Hisashi Sawada.

**Data curation:** Sohei Ito, Jeff Z. Chen, Hisashi Sawada.

**Formal analysis:** Sohei Ito, Jeff Z. Chen, Hisashi Sawada.

**Funding acquisition:** Alan Daugherty, Hisashi Sawada.

**Investigation:** Deborah A. Howatt, Jessica J. Moorleghen.

**Validation:** Hisashi Sawada.

**Writing – original draft:** Hisashi Sawada.

**Writing – review & editing:** Sohei Ito, Hong S. Lu, Alan Daugherty, Hisashi Sawada.

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
