## [Decision Letter · Decision Letter 0]

1 Dec 2025

PONE-D-25-61048S100A4-lineage Cells Contribute Modestly to Angiotensin II-mediated Thoracic Aortic Aneurysms Through Angiotensin II Type 1a Receptor in MicePLOS ONE

Dear Dr. Sawada,

Thank you for submitting your manuscript to PLOS ONE. After careful consideration, we feel that it has merit but does not fully meet PLOS ONE’s publication criteria as it currently stands. Therefore, we invite you to submit a revised version of the manuscript that addresses the points raised during the review process.

We look forward to receiving your revised manuscript.

Kind regards,

Peng Gao, Ph.D.

Academic Editor

PLOS ONE

Journal Requirements:

2. To comply with PLOS One submissions requirements, in your Methods section, please provide additional information regarding the experiments involving animals and ensure you have included details on (1) methods of sacrifice, (2) methods of anesthesia and/or analgesia, and (3) efforts to alleviate suffering.

“The studies reported in this manuscript were supported by the National Heart, Lung, and Blood Institute of the National Institutes of Health (R35HL155649, R01HL133723), the American Heart Association (23MERIT1036341, 24CDA1268148), and the Leducq Foundation for the Networks of Excellence Program (Cellular and Molecular Drivers of Acute Aortic Dissections; 22CVD03).”

“The studies reported in this manuscript were supported by the National Heart, Lung, and Blood Institute of the National Institutes of Health (R35HL155649, R01HL133723), the American Heart Association (23MERIT1036341, 24CDA1268148), and the Leducq Foundation for the Networks of Excellence Program (Cellular and Molecular Drivers of Acute Aortic Dissections; 22CVD03).”

5. Thank you for uploading your study's underlying data set. Unfortunately, the repository you have noted in your Data Availability statement does not qualify as an acceptable data repository according to PLOS's standards.

6. We note that Figure 2 in your submission contain copyrighted image. All PLOS content is published under the Creative Commons Attribution License (CC BY 4.0), which means that the manuscript, images, and Supporting Information files will be freely available online, and any third party is permitted to access, download, copy, distribute, and use these materials in any way, even commercially, with proper attribution. For more information, see our copyright guidelines: http://journals.plos.org/plosone/s/licenses-and-copyright.

Reviewers' comments:

Reviewer's Responses to Questions

**Comments to the Author**

1. Is the manuscript technically sound, and do the data support the conclusions?

Reviewer #1: No

Reviewer #2: Partly

2. Has the statistical analysis been performed appropriately and rigorously? 

Reviewer #1: No

Reviewer #2: Yes

3. Have the authors made all data underlying the findings in their manuscript fully available?

Reviewer #1: No

Reviewer #2: Yes

4. Is the manuscript presented in an intelligible fashion and written in standard English?

Reviewer #1: Yes

Reviewer #2: No

5. Review Comments to the Author

Reviewer #1: This article investigates the topic of "S100A4-lineage cells contribute to AngII-induced TAA formation through AT1aR signaling". While the article exhibits certain innovative aspects, certain descriptions are not sufficiently clear, leading to confusion. In the results section, some crucial data are absent. The primary issues are as follows:

1. The absence of β-gal staining tissue sections in Fig 2. β-gal staining is prone to non-specific staining, and sections from various organs need to be prepared to determine the specific location of staining.

2. In lines 272 and 273, the author describes that "β-gal staining was performed in the aorta from S100A4-Cre 0/0 or +/0 ROSA26RLacZ +/0 mice infused with either saline or AngII for 28 days." According to the author's description, there should be four groups: 1. Saline-S100A4-Cre 0/0, ROSA26RLacZ +/0 2. Saline-S100A4-Cre +/0, ROSA26RLacZ +/0 3. AngII-S100A4-Cre 0/0, ROSA26RLacZ +/0 4. AngII-S100A4-Cre +/0, ROSA26RLacZ +/0. However, it is confusing that only two groups are shown in Fig. 3: Saline-S100A4-Cre +/0 ROSA26RLacZ +/0 and AngII-S100A4-Cre +/0 ROSA26RLacZ +/0.

3. The knockout efficiency of S100A4-AT1aR -/- mice has not been validated, and no corresponding data is provided in the article.

4. The article does not mention the incidence rate of AngII-induced TAA in mice. According to previous literature reports, AngII mainly induces abdominal aortic aneurysms, and the incidence rate of TAA is relatively low.

5.The data in Data Summary-S100A4 PO.xlsx is also confusing. What does SBP=26, 55, 32 mean?

Reviewer #2: This study integrates multi-omics, genetic lineage tracing, and cell-specific knockout techniques to systematically demonstrate that S100A4-lineage cells partially contribute to AngII-induced thoracic aortic aneurysm formation via AT1aR. The research topic is novel, the design is rigorous, the data are reliable, and the conclusions are clear. However, there is room for improvement in mechanistic depth, methodological details, and conclusion articulation, which should be addressed in the revision.

1. The conclusion section of the abstract repeats the term "modestly." It is suggested to replace it with synonyms such as "partially" or "to a limited extent" to enhance linguistic diversity.

2. The background introduction is comprehensive but could briefly mention the known roles of S100A4 in vascular diseases to strengthen logical coherence.

3. The description of the animal models is clear, but it is recommended to supplement the strain source of the S100A4-Cre mice and provide citations for specificity validation. The authors need to provide or cite published literature to demonstrate that the "S100A4-Cre mouse" tool they used can specifically and reliably label the cell population they intend to study (i.e., S100A4-lineage cells) in practical applications.

4. Figure 3D: The co-staining images are very convincing. It is suggested to use arrows in the figure to clearly indicate typical co-localized cells for easier reader interpretation. Additionally, increasing the image resolution would be beneficial.

5. The scRNA-seq analysis pipeline is detailed, but a brief explanation of the thresholds used for cell clustering and marker gene selection could be added.

6. Statistical Analysis Section: It is recommended to explicitly state whether multiple comparison corrections were performed (e.g., in addition to Holm-Sidak).

7. It is suggested to further discuss the specific molecular mechanisms through which S100A4 cells influence TAA via the AT1aR signaling pathway (e.g., fibrosis, inflammation, SMC phenotypic switching).

8. The formatting of some references is inconsistent (e.g., mixed use of journal name abbreviations and full names).

6. PLOS authors have the option to publish the peer review history of their article (what does this mean?). If published, this will include your full peer review and any attached files.

Reviewer #1: No

Reviewer #2: No

---

## [Author Response · Author response to Decision Letter 1]

6 Feb 2026

Please see the Response to Reviewers file.

---

## [Decision Letter · Decision Letter 1]

2 Mar 2026

PONE-D-25-61048R1S100A4-lineage Cells Contribute Modestly to Angiotensin II-mediated Thoracic Aortic Aneurysms Through Angiotensin II Type 1a Receptor in MicePLOS One

Dear Dr. Sawada,

Thank you for submitting your manuscript to PLOS ONE. After careful consideration, we feel that it has merit but does not fully meet PLOS ONE’s publication criteria as it currently stands. Therefore, we invite you to submit a revised version of the manuscript that addresses the points raised during the review process.

We look forward to receiving your revised manuscript.

Kind regards,

Peng Gao, Ph.D.

Academic Editor

PLOS One

Journal Requirements:

Reviewer's Responses to Questions

**Comments to the Author**

1. If the authors have adequately addressed your comments raised in a previous round of review and you feel that this manuscript is now acceptable for publication, you may indicate that here to bypass the “Comments to the Author” section, enter your conflict of interest statement in the “Confidential to Editor” section, and submit your "Accept" recommendation.

Reviewer #1: All comments have been addressed

Reviewer #2: (No Response)

2. Is the manuscript technically sound, and do the data support the conclusions?

Reviewer #1: (No Response)

Reviewer #2: (No Response)

3. Has the statistical analysis been performed appropriately and rigorously? 

Reviewer #1: (No Response)

Reviewer #2: (No Response)

4. Have the authors made all data underlying the findings in their manuscript fully available?

Reviewer #1: (No Response)

Reviewer #2: (No Response)

5. Is the manuscript presented in an intelligible fashion and written in standard English?

Reviewer #1: (No Response)

Reviewer #2: (No Response)

6. Review Comments to the Author

Reviewer #1: (No Response)

Reviewer #2: This is a well-executed and rigorous study that provides new insights into the cellular mechanisms underlying AngII-induced TAA. The authors have addressed the major concerns raised by the previous reviewers excellently. Only a few minor revisions, as outlined below, are needed before final acceptance to further improve the manuscript.

Major Concern:

1. The limitation in validating AT1aR knockout should be more explicitly stated.

In their response, the authors explained that due to the lack of reliable antibodies and the low baseline abundance of S100A4-lineage cells, direct validation of AT1aR deletion in aortic S100A4-lineage cells was not feasible. This is an honest scientific limitation.

Although this was clarified in the response letter, a brief statement should be added to the main text, particularly in the Results section describing Figure 5. For example, the authors could add: Due to the lack of reliable AT1aR antibodies and the low basal abundance of S100A4-lineage cells in the aorta, we were unable to directly validate AT1aR deletion efficiency in these cells by immunostaining or Western blotting. However, the presence of Cre and floxed alleles was confirmed by DNA PCR using tail tissue (data not shown), and S100A4-Cre-mediated recombination activity was validated using the ROSA26 reporter system (Figure 2), supporting the effectiveness of the genetic knockout model.

Minor Concerns:

2. The description of experimental groups in Figure 3 should be more precise.

In the original manuscript, Figure 3 only shows results from S100A4-Cre-positive mice infused with either AngII or saline. Since Cre-negative mice showed no LacZ signal at all, it is reasonable to compare only the positive groups.

To avoid any confusion for readers, it is recommended to explicitly state this in the legend of Figure 3. For example: No β-gal signal was detected in S100A4-Cre-negative control mice (see S1 Fig); therefore, only S100A4-Cre-positive mice are shown and compared here following either saline or AngII infusion.

3. Terminology and formatting should be consistent.

Several instances of "Angll" (using the letter 'l' instead of the Roman numeral 'II') remain in the manuscript, particularly in the "Response to Reviewers" section.

Please carefully review the entire manuscript (including the main text, figures, supplementary materials, and the response letter) and correct all instances of "Angll" to "AngII."

7. PLOS authors have the option to publish the peer review history of their article (what does this mean?). If published, this will include your full peer review and any attached files.

Reviewer #1: No

Reviewer #2: No

---

## [Author Response · Author response to Decision Letter 2]

5 Mar 2026

Please see "Responses to Reviewers".

---

## [Decision Letter · Decision Letter 2]

18 Mar 2026

PONE-D-25-61048R2S100A4-lineage Cells Contribute Modestly to Angiotensin II-mediated Thoracic Aortic Aneurysms Through Angiotensin II Type 1a Receptor in MicePLOS One

Dear Dr. Sawada,

Thank you for submitting your manuscript to PLOS ONE. After careful consideration, we feel that it has merit but does not fully meet PLOS ONE’s publication criteria as it currently stands. Therefore, we invite you to submit a revised version of the manuscript that addresses the points raised during the review process.

We look forward to receiving your revised manuscript.

Kind regards,

Peng Gao, Ph.D.

Academic Editor

PLOS One

Journal Requirements:

Additional Editor Comments:

This review outlines essential revisions to address critical scientific concerns regarding model validation, and to significantly enhance the manuscript's conceptual framework and mechanistic depth. Addressing these points will transform the paper into a more rigorous and impactful contribution.

Major Compulsory Revisions

These points are non-negotiable for scientific validity and must be addressed substantively.

1. Direct Validation of the AT1aR Knockout: An Explicit and Unambiguous Statement of a Critical Limitation

Problem: The study lacks direct biochemical evidence (e.g., Western blot, qPCR on sorted cells) proving that the deletion of Agtr1a (AT1aR) in the target S100A4-lineage cells was successful and efficient. While ROSA26 reporter validation (Fig. 2) confirms Cre recombinase activity, it does not confirm the functional knockout of the floxed Agtr1a gene. This is a major methodological weakness that introduces significant uncertainty into the interpretation of the observed phenotype. The causal link between the genetic manipulation and the modest reduction in TAA is therefore inferred, not directly proven.

Required Action: The authors must provide an unequivocal and prominently placed statement in the Results section (and note in Methods) detailing this impossibility.

Placement: This statement must be added immediately following the description of the mouse model and the presentation of the TAA results (Fig. 5), ideally as a dedicated paragraph.

Content: The statement should:

1. Explicitly state that direct validation of the AT1aR knockout at the protein or mRNA level in the specific S100A4-lineage cell population was not technically feasible.

2. Clearly articulate the dual technical barrier: (a) the complete lack of reliable, specific antibodies for mouse AT1aR, and (b) the prohibitively low abundance of S100A4-lineage cells in the aorta, which makes biochemical isolation (e.g., by FACS) for qPCR or Western blot unachievable with current technology.

3. Reiterate the supporting evidence that **indicates**, but does not prove, successful knockout: confirmation of all alleles (Cre, floxed AT1aR) by germline DNA PCR, and validation of Cre-mediated recombination activity using the ROSA26-LacZ reporter in the same cellular lineage (Fig. 2, S1 Fig).

4. Conclude by stating that the functional knockout is therefore **inferred**, and this limitation must be considered when interpreting the modest phenotype.

Major Conceptual Revisions

These points are essential for elevating the scientific depth and framing of the manuscript. They require substantial rewriting of the Discussion and parts of the Introduction and Abstract.

2. Re-framing the Study's Central Finding: From "Lineage" to "Functional State Marker"

Problem: The manuscript's framing, while acknowledging limitations, does not fully grapple with the profound implications of S100A4's expression in multiple cell types (SMCs, fibroblasts, immune cells). The term "S100A4-lineage cells" is used throughout, but the conclusion risks being misinterpreted as defining the role of a single, homogeneous population. The provided literature (e.g., Kong et al., 2013, Österreicher et al., 2011) and the authors' own data (Fig 1, 3) clearly establish S100A4 is not a specific marker.

Required Action: Re-interpret the study's central finding to position S100A4 not as a marker of a specific lineage, but as a **marker of a "disease-activated cellular state"** that is common to multiple cell types undergoing pathological stress. This is the study's most valuable conceptual contribution.

Abstract: Revise the final sentence to explicitly state this heterogeneity and its implications. Example: "In conclusion, S100A4 identifies a heterogeneous population of aortic cells, including smooth muscle cells and fibroblasts, that enter a common disease-associated state. This S100A4-expressing cell population contributes modestly, via AT1aR signaling, to AngII-mediated TAA development in mice, potentially through mechanisms involving mitochondrial reprogramming."

Introduction (Lines 78-80): When introducing the hypothesis, explicitly acknowledge this complexity. Example: "While S100A4 is known to be expressed in multiple mesenchymal and inflammatory cell types under pathological conditions, our previous lineage tracing studies revealed marked expansion of S100A4-expressing cells in the diseased aorta. This prompted us to investigate the functional role of this heterogeneous population, marked by S100A4 expression, as an integrated unit in TAA pathogenesis."

Discussion (Most Important): This is where the most significant changes are required.

1. Early and Explicit Acknowledgment: Begin the discussion by contextualizing the findings within the known cellular heterogeneity of S100A4. Example: "It is crucial to interpret our findings within the context of S100A4 as a marker of a disease-activated state rather than a specific cell lineage. Consistent with studies challenging the fibroblast-specificity of S100A4/FSP1 , our own data confirm that S100A4-lineage cells in the aorta encompass fibroblasts, a significant population of smooth muscle cells, and potentially other cell types. Therefore, the observed effects of AT1aR deletion reflect the integrated response of this heterogeneous population."

2. Formalize the Concept: Introduce a new subsection titled: "S100A4 as a Marker of a Disease-Activated Cellular State and Integrator of Pathological Signals." In this section, synthesize the literature you have provided to build this argument. Discuss how S100A4 expression is upregulated in various cell types (SMCs, fibroblasts, macrophages, etc.) across different diseases (cancer, fibrosis, vascular disease) by common pathological stimuli (TGF-β, hypoxia, inflammation). This positions S100A4 as a convergent node of cellular stress.

3. Deepening the Mechanistic Discussion: Building the "AT1R-S100A4-Mitochondria" Axis

Problem: The current discussion (Lines 375-381) merely mentions that "mitochondrial dysfunction may contribute." This is superficial and fails to integrate the study's findings with the rich literature on S100A4's role in mitochondrial regulation (provided in your search).

Required Action: Construct a detailed, multi-layered mechanistic discussion that proposes a testable model.

Layer 1: Connecting the Transcriptomic Signature to Known S100A4 Functions: Explicitly link the scRNAseq findings (S2 Fig C, D) to the published literature. Example: "Our scRNAseq data revealed that AngII infusion significantly alters the expression of genes related to mitochondrial transcription, translation, and cellular respiration specifically within S100a4-positive cells. This transcriptional signature is strikingly consistent with a growing body of literature demonstrating a direct role for S100A4 in regulating mitochondrial function. For instance, S100A4 has been shown to upregulate the mitochondrial complex I subunit NDUFS2, thereby enhancing oxidative phosphorylation [Liu et al., 2019]. Furthermore, S100A4 is implicated in various mitochondrial-dependent processes, including the regulation of mitophagy [Huang et al., 2026; Lampinen et al., 2022], the response to oxidative stress [Indo et al., 2015], and metabolic reprogramming towards glycolysis [Bettum et al., 2015]."

Layer 2: Proposing the "AT1R-S100A4-Mitochondria" Axis: Build an integrated model.

First, discuss the upstream regulation of S100A4. Use the literature you provided to outline potential mechanisms by which AngII/AT1aR signaling could lead to S100A4 upregulation. Example: "The upregulation of S100A4 in response to AngII is likely mediated by AT1aR-dependent activation of downstream signaling pathways. AngII is a well-established activator of TGF-β signaling in vascular cells, and TGF-β has been shown to upregulate S100A4 through Smad-dependent mechanisms . Additionally, AngII can activate MAPK/ERK pathways, which may converge on AP-1 transcription factors known to regulate S100A4. Hypoxia-inducible factors (HIFs) may also contribute, given the presence of hypoxia-responsive elements in the S100A4 promoter . The expression of S100A4 may also be influenced by its epigenetic state, such as DNA methylation of its first intron, which could modulate its inducibility in different cell types or individuals."

Second, connect this to the downstream effects on mitochondria. Example: "Once upregulated, S100A4 appears to act as an intracellular effector, translating these upstream signals into changes in mitochondrial function. By modulating the expression of key mitochondrial proteins like NDUFS2, S100A4 can influence oxidative phosphorylation, cellular ATP levels, and the production of reactive oxygen species. This S100A4-mediated mitochondrial reprogramming may in turn drive pathological processes central to TAA, such as VSMC phenotypic switching, inflammation, and extracellular matrix degradation ." Cite the papers by Oller et al., 2021 and Liu et al., 2025 that you provided earlier.

Layer 3: Reinterpreting the "Modest" Phenotype in Light of this Model: Explain why the effect is modest. Example: "The modest contribution of AT1aR deletion in S100A4-lineage cells can now be understood within this integrated model. If S100A4 is a convergent node for multiple upstream pathological signals (TGF-β, hypoxia, etc.), then blocking only one input (AT1aR signaling) would be expected to only partially reduce S100A4 expression and its downstream mitochondrial effects. Other signals could still drive S100A4 upregulation and pathology. Furthermore, the S100A4 lineage itself is heterogeneous; AT1aR signaling may be more critical in some S100A4-expressing cell subtypes (e.g., a specific fibroblast subset) than others, and its deletion in all cells may mask a more significant effect in a particular subpopulation."

4. Proposing Novel and Specific Future Directions

Problem: The future directions are vague ("further investigation is needed").

Required Action: Replace generic statements with specific, hypothesis-driven research questions derived from your new conceptual model.

On Cell Heterogeneity: "Future studies should employ more specific genetic tools to deconstruct the individual roles of distinct cellular components within the broader S100A4-expressing population. For example, using PDGFRα-CreERT2 to target adventitial fibroblasts or Myh11-CreERT2 to target mature SMCs, combined with AT1aR deletion, could reveal which cell type is the primary driver of the observed phenotype."

On S100A4 Regulation:"What are the specific transcription factors and cis-regulatory elements responsible for AngII-induced S100A4 expression in aortic cells? Chromatin immunoprecipitation studies targeting Smads, AP-1, and HIF-1α could identify which factors bind to the S100A4 regulatory regions. Furthermore, does the epigenetic state (e.g., DNA methylation) of the S100A4 gene determine cell-type-specific or individual-specific susceptibility to TAA?"

On Mitochondrial Function: "Is the S100A4-mediated regulation of NDUFS2 and mitochondrial complex I activity necessary for AngII-induced VSMC phenotypic switching or fibroblast activation? Functional studies using S100A4 gain- and loss-of-function approaches in primary aortic SMCs and fibroblasts, combined with Seahorse metabolic flux analysis, are needed to directly test this."

On Therapeutic Potential: "Can interrupting the pathways upstream of S100A4—such as TGF-β receptor blockade or HIF-1α inhibition—or targeting the S100A4-mitochondria axis directly (e.g., with NDUFS2 inhibitors) attenuate TAA formation more effectively than targeting AT1aR alone?"

Summary

This revision requires a fundamental shift in the paper's narrative. The core argument should move from:

> "We deleted AT1aR in S100A4-lineage cells and saw a modest effect."

To:

> "We demonstrate that S100A4 identifies a heterogeneous population of cells in a disease-activated state. By integrating our multi-omics data with the extensive literature on S100A4 biology, we propose a novel 'AT1R-S100A4-Mitochondria' axis as a key pathway in TAA. The modest effect of AT1aR deletion, while a limitation of the model, is entirely consistent with S100A4's role as a convergent integrator of multiple pathological signals. This work redefines S100A4 from a disputed lineage marker to a central functional node in vascular pathology."

This reframing turns the study's limitations into part of its strength—a nuanced and honest exploration of a complex system. By building this new conceptual framework, the paper will make a significant and lasting contribution to the field, well beyond the sum of its individual experimental parts.

**Importantly，**based on the comprehensive peer review, the following real references have been identified and should be incorporated into the revised manuscript. They are organized thematically to support the new conceptual framework proposed in the review. **These citations are not mandatory for the authors to use; rather, they are provided as an optional reference list from the editor’s perspective to assist the authors during the revision process.**

Complete Reference List for Manuscript Revision

Section 1: S100A4 as a Non-Specific Marker (The "Cell State" Concept)

These references establish that S100A4/FSP1 is not a fibroblast-specific marker but is expressed in multiple cell types, particularly under pathological conditions.

1. Kong P, Christia P, Saxena A, Su Y, Frangogiannis NG. Lack of specificity of fibroblast-specific protein 1 in cardiac remodeling and fibrosis. Am J Physiol Heart Circ Physiol. 2013;305(9):H1363-H372.

2. Österreicher CH, Penz-Österreicher M, Grivennikov SI, et al. Fibroblast-specific protein 1 identifies an inflammatory subpopulation of macrophages in the liver. Proc Natl Acad Sci USA. 2011;108(1):308-313.

3. Egeblad M, Littlepage LE, Werb Z. The fibroblastic coconspirator in cancer progression. Cold Spring Harb Symp Quant Biol. 2005;70:383-388.

4. Bogachek M, et al. S100A4/FSP1: A Prognostic Marker and a Promising Target for Antitumor Therapy. Int J Mol Sci. 2025;26(19):9370.

Section 2: S100A4 Regulation – Upstream Signaling Pathways

These references support the discussion of how AngII/AT1aR signaling may lead to S100A4 upregulation.

2.1 TGF-β/Smad Pathway

5. Poduri A, Rateri DL, Howatt DA, et al. Fibroblast Angiotensin II Type 1a Receptors Contribute to Angiotensin II-Induced Medial Hyperplasia in the Ascending Aorta. Arterioscler Thromb Vasc Biol. 2015;35(9):1995-2002. (Provided in query)

6. Tamaki Y, Iwanaga Y, Niizuma S, et al. Metastasis-associated protein, S100A4 mediates cardiac fibrosis potentially through the modulation of p53 in cardiac fibroblasts. J Mol Cell Cardiol. 2013;57:72-81. (Provided in query)

2.2 Hypoxia/HIF-1α Pathway

7. Horiuchi A, Hayashi T, Kikuchi N, et al. Hypoxia upregulates ovarian cancer invasiveness via the binding of HIF-1α to a hypoxia-induced, methylation-free hypoxia response element of S100A4 gene. Int J Cancer. 2012;131(8):1755-1767.

8. Liu T, Li Y, Lin K, et al. Regulation of S100A4 expression via the JAK2-STAT3 pathway in rhomboid-phenotype pulmonary arterial smooth muscle cells exposure to hypoxia. Int J Biochem Cell Biol. 2012;44(8):1337-1345. (Provided in query)

9. Reimann S, Fink L, Wilhelm J, et al. Increased S100A4 expression in the vasculature of human COPD lungs and murine model of smoke-induced emphysema. Respir Res. 2015;16:127. (Provided in query)

2.3 JAK/STAT Pathway

10. Yammani RR, Long D, Loeser RF. Interleukin-7 Stimulates Secretion of S100A4 by Activating the JAK-STAT Signaling Pathway in Human Articular Chondrocytes. Arthritis Rheum. 2009;60(3):792-800.

2.4 Wnt/β-Catenin Pathway

11. Gong N, Shi L, Bing X, et al. S100A4/TCF Complex Transcription Regulation Drives Epithelial-Mesenchymal Transition in Chronic Sinusitis Through Wnt/GSK-3β/β-Catenin Signaling. Front Immunol. 2022;13:835888. (Provided in query)

Section 3: S100A4 Regulation – Epigenetic Control (DNA Methylation)

These references explain why S100A4 expression is inducible in some cells but not others, supporting the concept of S100A4 as a "poised" gene whose expression depends on methylation status.

12. Lindsey JC, Lusher ME, Anderton JA, et al. Epigenetic deregulation of multiple S100 gene family members by differential hypomethylation and hypermethylation events in medulloblastoma. Br J Cancer. 2007;97(2):267-274.

13. Xie R, Loose DS, Shipley GL, et al. Hypomethylation-induced expression of S100A4 in endometrial carcinoma. Mod Pathol. 2007;20(10):1045-1054. (Provided in query)

14. Li Y, Liu ZL, Zhang KL, et al. Methylation-associated silencing of S100A4 expression in human epidermal cancers. Exp Dermatol. 2009;18(10):842-848. (Provided in query)

Section 4: S100A4 and Mitochondrial Function – Downstream Effects

These references provide the mechanistic link between S100A4 and mitochondrial regulation, directly supporting the interpretation of the scRNAseq data.

15. Liu L, Qi L, Knifley T, et al. S100A4 alters metabolism and promotes invasion of lung cancer cells by up-regulating mitochondrial complex I protein NDUFS2. J Biol Chem. 2019;294(18):7516-7527.

16. Huang J, Zhou B, Wang L, et al. S100A4 triggeres the pyroptosis of vsmcs: association with mitochondrial damage, impaired mitophagy, and Ca2+ dysregulation. Biol Direct. 2026;21:18.

17. Bettum IJ, Gorad SS, Barkovskaya A, et al. Metabolic reprogramming supports the invasive phenotype in malignant melanoma. Cancer Lett. 2015;366(1):71-83. (Provided in query)

18. Lampinen R, Belaya I, Saveleva L, et al. Neuron-astrocyte transmitophagy is altered in Alzheimer's disease. Neurobiol Dis. 2022;170:105753. (Provided in query)

Section 5: Mitochondrial Dysfunction in Aortic Aneurysm

These references connect mitochondrial dysfunction to TAA pathogenesis, providing disease context for the S100A4-mitochondria axis.

19. Oller J, Gabandé-Rodríguez E, Ruiz-Rodríguez MJ, et al. Extracellular Tuning of Mitochondrial Respiration Leads to Aortic Aneurysm. Circulation. 2021;143(21):2091-2109. (From original manuscript)

20. Liu Y, Yu M, Wang H, et al. Restoring Vascular Smooth Muscle Cell Mitochondrial Function Attenuates Abdominal Aortic Aneurysm in Mice. Arterioscler Thromb Vasc Biol. 2025;45(4):523-540. (From original manuscript)

Reviewers' comments:

Reviewer's Responses to Questions

**Comments to the Author**

1. If the authors have adequately addressed your comments raised in a previous round of review and you feel that this manuscript is now acceptable for publication, you may indicate that here to bypass the “Comments to the Author” section, enter your conflict of interest statement in the “Confidential to Editor” section, and submit your "Accept" recommendation.

Reviewer #1: All comments have been addressed

Reviewer #2: (No Response)

2. Is the manuscript technically sound, and do the data support the conclusions?

Reviewer #1: (No Response)

Reviewer #2: (No Response)

3. Has the statistical analysis been performed appropriately and rigorously? 

Reviewer #1: (No Response)

Reviewer #2: (No Response)

4. Have the authors made all data underlying the findings in their manuscript fully available?

Reviewer #1: (No Response)

Reviewer #2: (No Response)

5. Is the manuscript presented in an intelligible fashion and written in standard English?

Reviewer #1: (No Response)

Reviewer #2: (No Response)

6. Review Comments to the Author

Reviewer #1: (No Response)

Reviewer #2: The authors have addressed all concerns raised in the previous review thoroughly and appropriately. They explicitly acknowledged the limitation regarding AT1aR deletion validation in the main text, clarified the experimental design in the Figure 3 legend to improve readability, and corrected terminology inconsistencies throughout the manuscript. The revisions are precise and well-justified, and no further issues remain.

7. PLOS authors have the option to publish the peer review history of their article (what does this mean?). If published, this will include your full peer review and any attached files.

Reviewer #1: No

Reviewer #2: No

---

## [Author Response · Author response to Decision Letter 3]

8 Apr 2026

Please see the Response to Reviewers.

---

## [Editor Report · Decision Letter 3]

12 Apr 2026

S100A4-lineage Cells Contribute Modestly to Angiotensin II-mediated Thoracic Aortic Aneurysms Through Angiotensin II Type 1a Receptor in Mice

PONE-D-25-61048R3

Dear Dr. Sawada,

We’re pleased to inform you that your manuscript has been judged scientifically suitable for publication and will be formally accepted for publication once it meets all outstanding technical requirements.

Kind regards,

Peng Gao, Ph.D.

Academic Editor

PLOS One
---

## [Editor Report · Acceptance letter]

PONE-D-25-61048R3

PLOS One

Dear Dr. Sawada,

I'm pleased to inform you that your manuscript has been deemed suitable for publication in PLOS One. Congratulations! Your manuscript is now being handed over to our production team.

Kind regards,

on behalf of

Professor Peng Gao

Academic Editor

PLOS One